# Associations Between MASLD, Ultra-Processed Food and a Mediterranean Dietary Pattern in Older Adults

**DOI:** 10.3390/nu17091415

**Published:** 2025-04-23

**Authors:** Isabella Commins, Daniel Clayton-Chubb, Jessica A. Fitzpatrick, Elena S. George, Hans G. Schneider, Aung Zaw Zaw Phyo, Ammar Majeed, Natasha Janko, Nicole Vaughan, Robyn L. Woods, Alice J. Owen, John J. McNeil, William W. Kemp, Stuart K. Roberts

**Affiliations:** 1Department of Gastroenterology, Alfred Health, Melbourne, VIC 3004, Australia; i.commins@alfred.org.au (I.C.); d.clayton-chubb@alfred.org.au (D.C.-C.); jessicafitzaptrick1@monash.edu (J.A.F.); a.majeed@alfred.org.au (A.M.); n.janko@alfred.org.au (N.J.); n.vaughan@alfred.org.au (N.V.); w.kemp@alfred.org.au (W.W.K.); 2School of Translational Medicine, Monash University, Melbourne, VIC 3004, Australia; 3Department of Gastroenterology, Eastern Health, Box Hill, VIC 3128, Australia; 4Department of Gastroenterology, St Vincent’s Hospital Melbourne, Fitzroy, VIC 3065, Australia; 5Institute for Physical Activity and Nutrition (IPAN), School of Exercise and Nutrition Sciences, Deakin University, Geelong, VIC 3220, Australia; elena.george@deakin.edu.au; 6School of Public Health and Preventive Medicine, Monash University, Melbourne, VIC 3004, Australia; h.schneider@alfred.org.au (H.G.S.); aungzawzaw.phyo@monash.edu (A.Z.Z.P.); robyn.woods@monash.edu (R.L.W.); alice.owen@monash.edu (A.J.O.); john.mcneil@monash.edu (J.J.M.); 7Department of Pathology, Alfred Health, Melbourne, VIC 3004, Australia; 8Department of Nutrition & Dietetics, Alfred Health, Melbourne, VIC 3004, Australia

**Keywords:** epidemiology, fatty liver, dietary patterns, healthy ageing, nutrition

## Abstract

**Background/Objectives:** Metabolic Dysfunction-Associated Steatotic Liver Disease (MASLD) is the most common liver disease worldwide, affecting 38% of the adult population globally. **Methods:** We examined the impact of the Mediterranean Diet and Ultra Processed Food (UPF) intake on the risk of prevalent MASLD in older adults. **Results:** Our major findings include that increased adherence to a Mediterranean Dietary pattern was associated with a decreased risk of MASLD. Additionally, we found that high UPF intake was associated with an increased risk of MASLD. Furthermore, our study found that even with a high UPF intake, the risk of MASLD decreased if the individual also had a higher Mediterranean Diet intake compared to a lower Mediterranean Diet intake. **Conclusions:** These results are of particular importance in older people, given the known links between MASLD, metabolic comorbidities and frailty. Public health messaging should focus on promoting Mediterranean dietary habits, and ways to help older people achieve this given the social and economic barriers they may face.

## 1. Introduction

Metabolic Dysfunction-Associated Steatotic Liver Disease (MASLD) is the most common liver disease worldwide, with an estimated global prevalence of 38% in adults, which is projected to reach 55% by 2040 [1]. In middle-aged people, MASLD confers a significant increase in morbidity and premature mortality via increased rates of cardiovascular disease (CVD) [2,3] as well as the progression of liver disease to cirrhosis and the development of hepatocellular carcinoma [4]. While the link between MASLD, mortality, and CVD is not as clear in older persons as it is in the middle-aged [5,6,7], there are known older person-specific associations including frailty [8] and persistent physical disability [9]. This is of particular concern given the world’s rapid transition towards an older population, with almost twice as many people surviving beyond 80 years now compared to 50 years ago [10,11]. An ageing population results in a greater number of individuals in our community living with chronic diseases, such as MASLD, as well as developing persistent disability and loss of independence [9,12]. These morbidities are a major concern for individuals but also have huge societal and economic impacts, placing a greater burden on our global healthcare system [13,14].

The morbidity associated with an ageing population is in part related to lifestyle factors, including dietary intake which is a well-recognised driver of health status [15]. However, the interplay between dietary patterns and age-related morbidity remains relatively understudied in older adults. It has been demonstrated that healthier dietary patterns—including adherence to the Mediterranean Diet—are associated with improved quality of life [16]. The Mediterranean Diet is a well-established dietary pattern, characterised by a high intake of olive oil, fruit, vegetables, nuts, wholegrains, legumes, and oily fish, and limited consumption of red and processed meat [17]. The benefits of adherence to a Mediterranean Diet in both the prevention and management of patients with MASLD has been well established [18,19], with both North American and European guidelines recommending it for dietary MASLD therapy. This is due to both its association with improvements in cardiovascular health as well as reductions in liver fat [20,21]. In addition, the Mediterranean Diet has been associated with reductions in all-cause mortality [22], CVD [23,24], cognitive impairment and dementia [25], and improved healthy ageing [26]. However, the unique physiological, psychosocial and economic factors that accompany ageing can act as significant barriers to maintaining and achieving a high quality diet in older adults [27].

The ultra-processed food (UPF) classification is a new way to assess ready-to-eat products which have emerged as detrimental to health, [28] and may have particular significance in older adults due to their convenience and accessibility [29]. As defined within the NOVA classification system [30], UPFs are classified as packaged foods that have undergone industrial processing with additives, and which are often provided in sophisticated packaging to create convenient, appealing and affordable products such as biscuits, pastries, confectionary, most breakfast cereals, flavoured yoghurts, many pre-made meals, muesli bars, protein powders, factory-made breads, crackers and soft drinks [31].

The relationship between UPF intake and steatotic liver disease in middle-aged adults is well established. Several studies have demonstrated the increased risk of developing MASLD, type two diabetes mellitus (T2DM) and the metabolic syndrome with increasing UPF intake [32,33,34]. Overall poorer diet quality, in addition to high saturated fat and refined grain intake seem to mediate this association between UPF and MASLD [35]. In addition, high UPF intake has been associated with higher rates of development of metabolic (dysfunction)-associated steatohepatitis (MASH)—the progressive form of steatotic liver disease—among people living with MASLD [36].

However, there remains a paucity of large studies exploring the relationship between dietary patterns and MASLD in older adults. Our group recently established dietary indices to quantify Mediterranean Diet and UPF intake [37] and found that low adherence to the Mediterranean Diet and increased UPF intake were associated with pre-frailty and frailty, highlighting the importance of dietary pattern assessment in this vulnerable population [32]. Using the same cohort, our present study aims to utilise these dietary indices to establish the relationship between dietary intake patterns of older Australians (via Mediterranean Diet adherence and UPF intake) on the risk of prevalent MASLD and its well-established co-morbidities.

## 2. Methods

### 2.1. Study Population

This is a post-hoc analysis of the ASPirin in Reducing Events in the Elderly (ASPREE) randomised trial [38] and the ASPREE Longitudinal Study of Older Persons (ALSOP) cohort study [39]. ALSOP collected data at trial baseline, and trial year 3—which was used for this study. Both baseline and outcomes trial data have been published previously [38,40,41,42] In brief, 16,703 Australians aged 70 years or older were recruited through primary care centres and randomised to 100 mg of aspirin or placebo to assess the impact of aspirin on incident CVD, mortality, and disability-free survival (amongst other key adjudicated endpoints). Key inclusion criteria include being free from clinically evident cardiovascular or cerebrovascular disease, self- or physician-reported dementia, having functional independence, and having a life expectancy of greater than five years. All participants provided written informed consent. The ASPREE and ALSOP studies were approved by local ethics committees, and ASPREE registered on ClinicalTrials.gov (NCT01038583) and the International Standard Randomised Controlled Trial Number Registry (ISRCTN83772183).

### 2.2. Sociodemographic Characteristics, Physical Examination Data, and Laboratory Data

During the ASPREE trial interviews, structured questionnaires and self-reported data were collected at baseline and during follow-up. This included information based on social and medical history, medication use, lifestyle factors (including alcohol intake and cigarette smoking), tests of cognitive function, and quality of life scores. Anthropometric measures were collected and recorded annually, including weight, height, abdominal circumference, blood pressure, and markers of physical function. Blood tests were also performed annually, including a fasting glucose and lipid profile. Additionally, as part of a separate sub-study—the ASPREE Healthy Ageing Biobank [43]—Australian participants were invited to provide serum for storage at both baseline and year 3. This has been used subsequently to determine (amongst others) liver function parameters including gamma-glutamyl transferase (GGT) values. The analysis of the Healthy Ageing Biobank serum was performed centrally at Alfred Health Pathology using an Abbott Alinity ci analyser and Abbott reagents (Abbott Diagnostics, Macquarie Park, NSW, Australia).

### 2.3. Identifying MASLD

We have previously identified a MASLD group in the ASPREE cohort at baseline [8] by utilising the Fatty Liver Index (FLI), a composite score based on Body Mass Index (BMI), abdominal circumference, serum triglycerides, and GGT [44]. This group was identified based on having their GGT from the Healthy Ageing Biobank [43] collected within 90 days of baseline data collection. For the purposes of this study, we utilised similar principles for GGT data from the Year 3 collection. However, this collection period was wider (i.e., covered more ASPREE data-collection time periods), and so to maximise the accuracy of the calculations we used the following algorithm (see also Figure 1A,B):1.If the GGT was taken within 90 days of data collection for the other FLI variables (BMI, abdominal circumference, and serum triglycerides), this data was used to calculate the FLI.2.If the GGT was taken more than 90 days from a data collection point, the points on either side of the GGT value were averaged (i.e., [BMI_TimePoint1_ + BMI_TimePoint2_]/2) and used.3.If one data point was missing when averaging, then the only data point available was used to calculate the FLI.

Following this, those with a FLI of ≥60 were classified as having hepatic steatosis. Individuals were then excluded if they met any of the MASLD exclusion criteria [45] (i.e., drank alcohol in excess of the pre-specified cut-offs of >14 drinks per week [females] or >21 drinks week [males], and/or took a steatogenic medication(s) including tamoxifen, methotrexate, glucocorticoids, or amiodarone). Similarly, if they did not meet one of the pre-specified markers of metabolic dysfunction inherent to the MASLD definition [45], they were also excluded. The data for these exclusion criteria were taken from the same time-points as the FLI calculation. For those with a FLI <60 (i.e., indeterminate or no steatosis), they were kept in the study assigned as no-MASLD comparators irrespective of alcohol or medication intake. A schematic to demonstrate this can be seen in Figure 1 and Figure 2A. If there was either no GGT data available or if the other anthropometric or triglyceride data was more than 1 year from the GGT collection, the FLI was not calculated, and the individual was not evaluated in this study.

### 2.4. Identifying Dietary Patterns

During the ASPREE clinical trial period, the ALSOP sub-study focusing on lifestyle factors commenced and was followed by further updated questionnaires at year 3 (the basis of this study) and year 5. Year 3 contained a 54-item food frequency questionnaire (FFQ) based on 12-month dietary recall, with detailed methodology described in previous publications [37,46] and with the scoring system and FFQ shown in the Appendix A. This FFQ was used to generate both a Mediterranean Diet adherence score (ASPREE-MDS) and UPF intake score (ASPREE-UPF) in the ALSOP population [37], which are used here to assess for the relationship between dietary intake and MASLD. In brief, the ASPREE-MDS was coded based on relative adherence to Mediterranean Diet Principles, where a higher score indicates greater adherence (maximum possible score 18). The ASPREE-UPF was based on the frequency and variety of UPFs consumed by the respondents, with higher scores indicating a larger quantity and/or variety of UPF intake (maximum possible score 25) with UPF being defined by NOVA classification system. This can be seen in brief in Figure 2B. The ASPREE FFQ does not include any data on portion size—therefore, caloric intake cannot be calculated. These scores are primarily built to be internally consistent within the population, rather than extrapolating to specific total or percentage energy values for individual nutrients or dietary components, comparing members of the population with one another. As such, analysis of these scores was conducted using tertiles, with the lowest tertile of each score as the reference value (i.e., the lowest tertile of UPF implies less UPF intake than the middle and top tertile; the lowest tertile of Mediterranean Diet Score implies the lowest adherence to these dietary principles). These tertiles were calculated only on the subpopulation with a FLI score (Figure 2C).

### 2.5. Statistical Approach

Data were compared using a one-way ANOVA or Kruskall-Wallis (for continuous variables) or a Chi-squared test (for categorical variables). When examining associations between both known covariates and the dietary scores in tertiles and their associations with the presence of MASLD, Poisson regression (with robust variance) was used to estimate prevalence relative risk ratios. The initial model utilised the dietary pattern tertiles in isolation, before additional covariate adjustments (including for age, sex, cardiometabolic comorbidities, educational attainment, and frailty [assessed by a deficit accumulation frailty index [47]]) were included. The deficit accumulation frailty index is a 67-item score generated yearly in ASPREE, based on the principles of Rockwood et al. [48]. More details can be seen in the derivation and validation paper published in 2022 [47]. Direct components of the FLI (BMI and abdominal circumference) were not included in any of the regression models. Subsequently, univariate relationships between the ASPREE-UPF and ASPREE-MDS tertiles with the individual components of the FLI (BMI, abdominal circumference, serum triglycerides, and GGT) were evaluated using linear regression. A *p* < 0.05 was considered statistically significant. Analyses were performed using Stata software v17.0 (Stata Corp LLC, College Station, TX, USA).

## 3. Results

### 3.1. Final Study Population

A total of 6753 participants still living at home had both a FLI classifiable using MASLD criteria and valid dietary scores using Year 3 ASPREE (BMI, abdominal circumference, serum triglycerides), ALSOP (dietary scores), and biochemistry (GGT) data.

The median (IQR) age of all participants was 76.74 (74.57–80.02), 45% were male and 98.8% were Caucasian. The mean BMI (±SD) of the cohort was 27.28 ± 4.36 kg/m^2^, 37.4% were overweight or obese. General characteristics of the participants stratified by FLI can be seen in Table 1 (comparing no-MASLD [FLI  < 30], indeterminate MASLD [FLI 30–60) and MASLD [FLI ≥ 60]). The median (IQR) ASPREE-MDS was 11.10 (9.98–12.68), and the median (IQR) UPF score was 6.06 (5.00–7.25).

### 3.2. MASLD and the Mediterranean Diet

A Poisson regression model with robust variance was constructed to examine the effect of both dietary pattern intakes on the risk of MASLD (FLI ≥ 60) compared with no-MASLD (FLI  < 30) (Table 2, Table 3 and Table 4). Compared to those in the lowest tertile of Mediterranean Diet Scores (ASPREE-MDS), participants with in the higher tertiles of ASPREE-MDS–i.e., those with greater adherence to a Mediterranean Diet pattern–were at a significantly lower risk of having MASLD (Q2 0.86 [0.79–0.92], *p* < 0.001 and Q3 0.71 [0.65–0.77], *p* < 0.001) (Table 2). When adjusted for age and sex, higher ASPREE-MDS continued to be associated with a lower risk of MASLD (Q2 0.88 [0.82–0.95], *p* = 0.001, Q3 0.74 [0.68–0.80], *p* < 0.001). When fully adjusting for variables of interest (diabetes, hypertension, chronic kidney disease, frailty and education level), higher ASPREE-MDS continued to carry a lower risk of MASLD (Q3 0.90 [0.83–0.97], *p* = 0.006].

### 3.3. MASLD and Ultra-Processed Foods

Participants with higher UPF intake were at a significantly higher risk of having MASLD (Q2 1.16 [1.07–1.27], *p* < 0.001 and Q3 1.25 [1.16–1.36], *p* < 0.001) (Table 3). When adjusted for age and sex, higher UPF intake continued to be associated with a higher risk of MASLD (Q2 1.13 [1.04–1.23], *p* = 0.003, Q3 1.16 [1.07–1.26], *p* < 0.001). When adjusting for the same key variables of interest, higher UPF intake continued to carry a higher risk of MASLD (Q2 1.12 [1.04–1.21], *p* = 0.002 and Q3 1.13 [1.05–1.22], *p* = 0.001).

### 3.4. MASLD and Both Dietary Scores

When incorporating both diet scores (ASPREE-MDS and ASPREE-UPF) into the same Poisson regression model, the results were similar to the analysis of individual dietary scores (Table 4). A greater adherence to the Mediterranean Diet led to a significantly lower risk of MASLD, but only for those participants with the highest ASPREE-MDS (Q3 0.88 [0.82–0.96] *p* = 0.002). Like our previous results, a higher intake of UPFs was associated with a higher risk of MASLD including when adjusting for UPF tertiles as well as known MASLD risk factors (Q2 1.13 [1.05–1.21], *p* = 0.001, Q3 1.16 [1.06–1.23], *p* = 0.001).

### 3.5. Sensitivity Analysis with Alternative FLI Cut-Off

When adjusting the same model to include FLI <60 (grouping participants with indeterminate and no-MASLD), the results were similar. Participants with highest ASPREE-MDS (Q3) were at a higher risk of having MASLD (0.90 [0.81–0.97] *p* = 0.014) even when adjusted for age, sex and other co-morbidities (Table 5). Similarly, participants with higher UPF intake were at a higher risk of MASLD (Q2 1.17 [1.07–1.27], *p* = 0.001 and Q3 1.18 [1.08–1.29], *p* < 0.001) when adjusted for all variables (Table 6).

Additionally, in an exploratory analysis of the unadjusted relationship between those with MASLD versus those with indeterminate and no-MASLD (FLI ≥ 60 vs. FLI < 60), the presence of MASLD increased with higher UPF intake and decreased with higher Mediterranean Diet adherence (Table 7).

### 3.6. Relationship Between Dietary Scores and FLI Components

A linear regression was performed to examine each individual component of the FLI score, and the impact that both diets had on these variables. A higher ASPREE-MDS was associated with lower triglycerides (Q2 −6.30 [−9.29–−3.30], *p* < 0.001 and Q3 −10.27 [−13.30–−7.24], *p* < 0.001), lower BMI (Q3 −0.72 [−0.97–−0.46], *p* < 0.001) and a lower waist circumference (Q2 −0.74 [−1.40–−0.08], *p* = 0.029 and Q3 −2.05 [−2.73–−2.39], *p* < 0.001). Higher Mediterranean Diet intake was not associated with a lower GGT (Table 8). A higher UPF intake was associated with higher triglycerides (Q2 4.47 [1.44–7.51], *p* = 0.004, Q3 4.21 [1.13–7.29], *p* = 0.007), higher BMI (Q2 0.37 [0.12–0.63], *p* = 0.004, Q3 0.36 [0.10–0.62], *p* = 0.006) and a higher waist circumference (Q2 0.95 [0.29–1.62] *p* = 0.005, Q3 1.07 [0.39–1.75], *p* = 0.002) (Table 9). Higher UPF intake was not associated with higher GGT.

## 4. Discussion

Our study looked at intake of both the Mediterranean Diet and Ultra-Processed Foods in older adults and found significant associations with the prevalence of steatotic liver disease. In this work focusing on relatively healthy community-dwelling older adults, we found that both a higher intake of the Mediterranean Diet and a lower intake of UPFs were associated with a decreased risk of having MASLD. Additionally, we found that individuals who had a higher adherence to the Mediterranean Diet, even with a concurrent high UPF intake, had a decreased presence of MASLD. These results contribute valuable knowledge to the existing literature through both providing initial data for future prospective studies as well as supporting some public health dietary recommendations for older persons.

The link between adherence to a Mediterranean Diet pattern and a lower risk of MASLD among middle-aged adults has been well recognised in the literature. Previous work has established that the Mediterranean Diet reduces liver steatosis, improves insulin sensitivity, reduces the severity of liver disease and reduces the likelihood of MASH in adults with MASLD [19,49,50]. However, there is a distinct paucity of evidence looking at the effect of the Mediterranean Diet on MASLD specifically in older adults. Our study has demonstrated that people who had a greater adherence to the Mediterranean Diet were at a significantly lower risk of having MASLD. When adjusting for age, sex and variables known to be associated with MASLD, such as diabetes, hypertension and chronic kidney disease, this finding remained consistent. In addition, we demonstrated that higher Mediterranean Diet intake was associated with lower triglycerides, lower BMI and a lower waist circumference in older adults.

Previous studies have also established the link between UPF intake and steatotic liver disease. In addition to an increased risk of developing T2DM and the metabolic syndrome with high UPF intake [32,33,34], one study used vibration-controlled transient elastography (VCTE) to demonstrate that high UPF consumption is positively associated with hepatic steatosis [51]. Moreover, high UPF intake has been associated with higher rates of development of MASH among persons living with MASLD [36]. Our study demonstrated that higher UPF intake was associated with a higher presence of MASLD. This association remained true even when adjusting for known variables that contribute to the development of MASLD. This relationship appears to be incremental in nature, i.e., the more UPF intake, the greater the risk of MASLD. Furthermore, higher UPF intake was associated with higher triglycerides, higher BMI and a higher waist circumference.

Interestingly, our study found that even with a high UPF intake, the risk of MASLD is decreased if the individual also has a higher MDS compared to a lower MDS (Table 7). This is a novel finding, as there is currently a lack of data in the literature comparing UPF and Mediterranean Diet directly, and their association with the risk of MASLD in older adults. This finding highlights the particular significance of the Mediterranean Diet and its potent anti-inflammatory and anti-oxidative effects [52,53], perhaps indicating a protective mechanism against MASLD in the presence of other lifestyle related risk factors such as UPFs. Mechanistically, there are multiple known phytochemicals found in our food which may influence the development of steatosis and ameliorate dyslipidemia, including isoflavones [54,55,56] (often found in legumes), the polyphenols found in extra virgin olive oil [57,58], and other bioactive compounds found in many of the plant-based foods commonly consumed in the Mediterranean Diet [59]. Similarly, the Mediterranean Diet may have beneficial effects on the gut microbiome [60], which is potentially causative (or protective) for the development of MASLD [61]. This protective effect of the Mediterranean Diet even in the context of higher UPF intake also highlights the importance of overall dietary quality and the context in which foods are consumed, rather than individual nutrients being considered in isolation. This is an area that warrants further research, as does evaluating whether particular classes of UPFs are at a greater risk of being associated with MASLD.

The dietary choices of older individuals in the community are influenced by numerous factors associated with ageing including financial stress, appetite and taste changes, social isolation, and poor mobility [16,62]. The concept of food insecurity, which is defined as the lack of consistent access to enough safe and nutritious food to maintain a healthy life, is increasingly being recognised as a factor that influences overall metabolic health [63,64]. In developed nations, food insecurity has been associated with MASLD, advanced liver fibrosis, and all-cause mortality, particularly in low-income adults [64,65,66]. Research has also shown that food insecurity is associated with low adherence to a Mediterranean Diet [67,68,69] as well as malnutrition in older adults [70]. In addition, a recent study found that in high socio-demographic index countries, food insecurity and the consumption of UPFs were associated with higher MASLD prevalence [71]. Our study confirms the well-established link between MASLD and UPFs, but more importantly, it highlights that the association is also strong in older people. As such, the convenience, longer shelf-life and inexpensive nature of UPFs may present as attractive dietary options for this population group. Social and economic drivers of UPF intake in older adults warrants further exploration to allow targeted public health messaging to convey the importance of healthy dietary patterns.

Our study has numerous strengths, including its large sample size which is uniquely composed of older adults, its rigorous and comprehensive data collection from recruitment through to follow up, its robust dietary and lifestyle questionnaires, its ability to identify those with MASLD, and thus subsequent capacity to draw robust associations between MASLD and dietary intake in older adults. We also only included older persons living at home (rather than in aged care facilities), likely reflecting their longer-term pattern of eating. There are, however, a few limitations to our study. The use of FLI to identify those with MASLD is a limitation in the diagnosis of steatotic liver disease rather than the gold-standard of liver biopsy or radiology. However, the FLI has been previously validated as an indicator of hepatic steatosis in large epidemiologic studies, including in a study with a very similar population group (predominantly older, Caucasian adults) to ours [72]. Additionally, while we used the dietary scores established in a similar study by our group, these scores have several limitations [37]. Given that the scores were derived from the ASPREE dietary questionnaire, and not a validated food frequency questionnaire, we did not have information on the quantities of food eaten, only the frequencies. This limited our ability to apply and compare with previously used epidemiological dietary scores, as well as limiting our understanding of whether there are threshold quantities of important dietary components for either benefit or risk or are related to total calorie intake. Furthermore, the ASPREE-dietary questionnaire was self-reported, based on the previous 12 months of food intake, potentially leading to recall bias. An additional limitation is that the ASPREE cohort represents a relatively healthy group of older adults, which may skew the data away from population norms of UPF intake, Mediterranean Diet adherence, and indeed the prevalence of MASLD. However, this is somewhat mitigated against by using the population as their own controls for both UPF intake and the MDS and represents a population of older adults still living at home who are most likely to be the target of dietary and lifestyle interventions. An additional limitation inherent to this study design is the lack of potential biomarker discovery to evaluate any specific protective factors found in the Mediterranean Diet for those also eating significant quantities of UPFs; future biomarker-driven mechanistic work would be valuable. Finally, this population was predominantly Caucasian, limiting the ability to apply these findings to other ethnicities.

## 5. Conclusions

In conclusion, higher adherence to a Mediterranean Diet and lower UPF intake are associated with a reduced risk of prevalent steatotic liver disease in older people, even when adjusting for variables known to be associated with MASLD. These results are of particular importance in older people, given the known links between MASLD and metabolic comorbidities and frailty [8]. Public health messaging should focus on promoting Mediterranean dietary habits, and ways to help older people achieve this given the social and economic barriers they may face.

## Figures and Tables

**Figure 1 nutrients-17-01415-f001:**
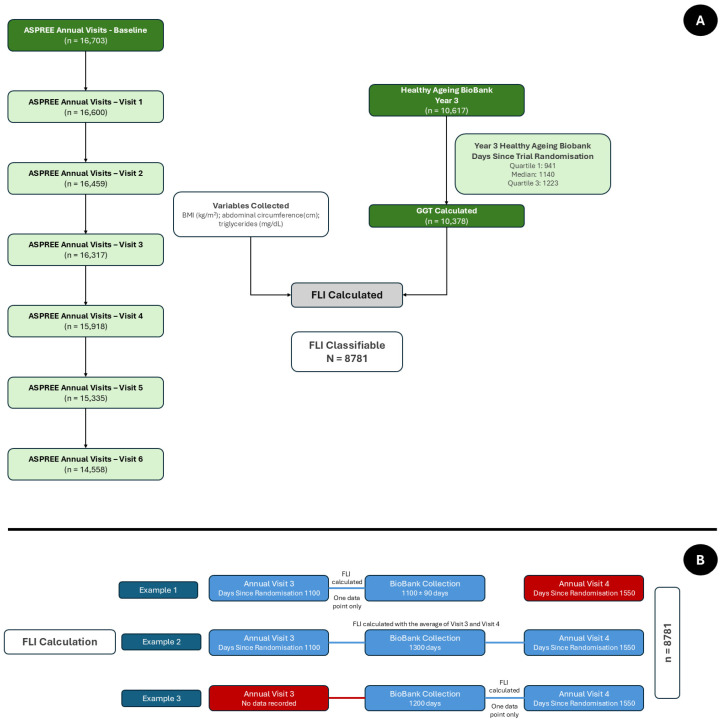
FLI Calculation Process. (**A**) Collection of FLI variables; (**B**) Example FLI calculation process.

**Figure 2 nutrients-17-01415-f002:**
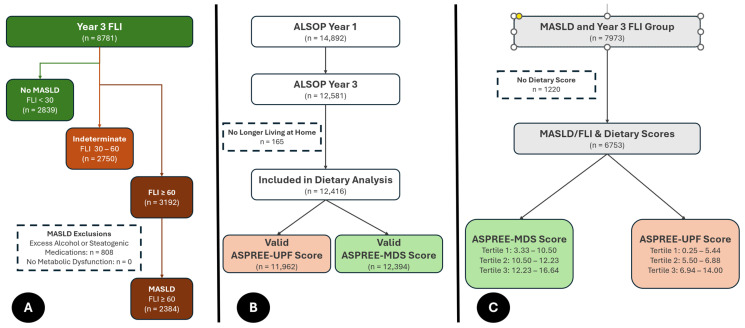
MASLD identification and Dietary Score Calculation. (**A**) Assignment of MASLD status from those with a calculable FLI; (**B**) Assignment of UPF and Mediterranean Dietary Scores; (**C**) Assignment of dietary score tertiles to those with/without MASLD.

**Table 1 nutrients-17-01415-t001:** Overall Population Data.

Characteristics	All Participants	FLI < 30	FLI 30–60	FLI ≥ 60	*p*-Value
Number of participants	6753	2454 (36.3%)	2336 (34.6%)	1963 (29.1%)	
Sex (male)	3050 (45%)	823 (33.5%)	1232 (52.7%)	985 (50.2%)	<0.001
Age (years) (Median [IQR])	76.74 (74.57–80.02)	76.87 (74.61–80.43)	76.87 (74.67–80.13)	76.45 (74.41–79.40)	<0.001
Caucasian ethnicity (n, %)	6674 (98.8%)	2421 (98.7%)	2308 (98.8%)	1945 (99.1%)	0.416
BMI (mean ± SD) kg/m^2^	27.28 ± 4.36	23.73 ± 2.34	27.19 ± 2.23	31.83 ± 4.02	<0.001
BMI Category ^a^ (n, %) kg/m^2^ Underweight Healthy weight Overweight Obese	981 (14.5%) 3247 (48.1%) 1856 (27.5%) 668 (9.9%)	917 (37.4%) 1465 (59.7%) 72 (2.9%) 0 (0.0%)	61 (2.6%) 1486 (63.6%) 765 (32.8%) 23 (1.0%)	3 (0.02%) 296 (15.1%) 1019 (51.9%) 645 (32.9%)	<0.001
Waist Circumference (mean ± SD) cm Large waist circumference ^b^ (n, %) cm	95.35 ± 12.18 3510 (52.0%)	84.68 ± 8.17 424 (17.3%)	96.31 ± 6.32 1298 (55.6%)	107.55 ± 9.35 1788 (91.1%)	<0.001 <0.001
**Laboratory Parameters**					
Glucose (mean ± SD) mg/dL	98.76 ± 17.55	94.01 ± 13.23	98.58 ± 17.12	104.91 ± 20.65	<0.001
Total Cholesterol (mean ± SD) mg/dL	197.79 ± 39.50	202.22 ± 37.02	197.07 ± 38.63	193.11 ± 39.56	0.006
HDL Cholesterol (mean ± SD) mg/dL	61.94 ± 17.84	70.86 ± 18.08	59.98 ± 15.58	53.12 ± 14.69	<0.001
LDL Cholesterol (mean ± SD) mg/dL	113.15 ± 33.97	114.19 ± 32.58	114.79 ± 33.93	109.90 ± 35.50	<0.001
Triglycerides (mean ± SD) mg/dL	113.27 ± 1.99	85.81 ± 30.20	111.11 ± 42.39	150.17 ± 60.93	<0.001
eGFR (mean ± SD) mL/min/1.73 m^2^	70.72 ± 13.85	72.67 ± 13.30	70.59 ± 13.60	68.46 ± 14.46	<0.001
**Cardiometabolic Conditions**					
Diabetes Mellitus ^c^ (n, %)	642 (9.5%)	102 (4.2%)	194 (8.3%)	346 (17.6%)	<0.001
Hypertension ^d^ (n, %)	4903 (72.6%)	1573 (64.1%)	1729 (74.0%)	1601 (81.6%)	<0.001
Chronic Kidney Disease ^e^ (n, %)	1903 (28.2%)	594 (24.2%)	622 (26.6%)	687 (35.0%)	<0.001
**Lifestyle Factors**					
Currently Smoking (n, %)	137 (2.0%)	59 (2.4%)	47 (2.0%)	31 (1.6%)	0.154
Currently Drinking Alcohol (n, %)	4981 (73.8%)	1850 (75.4%)	1772 (75.9%)	1359 (69.2%)	<0.001
**Dietary Scores**					
ASPREE-MDS (Median [IQR])	11.10 (9.98–12.68)	11.76 (10.29–12.96)	11.34 (9.89–12.59)	11.06 (9.65–12.42)	<0.001
ASPREE-UPF (Median [IQR])	6.06 (5–7.25)	6 (4.75–7.19)	6.09 (5–7.25)	6.25 (5.13–7.44)	<0.001
**Frailty**					
Deficit Accumulation Frailty Index [47] (n, %) Not Frail Pre-Frail Frail	3725 (55.2%) 2439 (36.1%) 583 (8.6%)	1685 (68.7%) 647 (26.4%) 119 (3.9%)	1388 (59.5%) 792 (33.9%) 153 (6.6%)	652 (33.2%) 1000 (50.9%) 311 (15.8%)	<0.001
**Education Completion**					
≤11 years (n, %)	3202 (47.4%)	1034 (42.1%)	1097 (47.0%)	1071 (54.6%)	<0.001
12 years (n, %)	749 (11.1%)	292 (11.9%)	251 (10.7%)	206 (10.5%)	
≥13 years (n, %)	2802 (41.5%)	1128 (46.0%)	988 (42.4%)	686 (34.9%)	
**Living Situation**					
Living alone (n, %)	2160 (32.0%)	850 (34.6%)	716 (30.7%)%)	594 (30.3%)	0.002

^a^ Underweight = BMI < 23.0 kg/m^2^; Healthy weight = BMI 23.0–28.0 kg/m^2^; Overweight = BMI 28.0–33.0 kg/m^2^; Obese = BMI > 33.0 kg/m^2^. ^b^ Large waist circumference = If Asian, ≥88 cm (males) and ≥80 cm (females); if non-Asian, ≥102 cm (males) and ≥90 cm (females). ^c^ Defined as one or more of (a) self-reported diabetes mellitus, (b) prescription of at least one glucose-lowering therapy at baseline, (c) fasting blood sugar of ≥7.0 mmol/L; ^d^ Defined as systolic blood pressure ≥130 mm Hg and/or prescription of at least one antihypertensive at baseline. ^e^ Chronic kidney disease is defined as defined as eGFR < 60 mL/min/1.73 m^2^ and/or albumin to creatinine ratio ≥ 3 mg/mmol.

**Table 2 nutrients-17-01415-t002:** MASLD + Mediterranean Diet (Poisson Regression).

MASLD (FLI ≥ 60) vs. No-MASLD (FLI < 30)						
	Unadjusted/Crude Model Relative Risk (95% CI)	*p*-Value	Age and Sex Adjusted Model Relative Risk (95% CI)	*p*-Value	Fully Adjusted Model Relative Risk (95% CI)	*p*-Value
Mediterranean Diet Score						
Q1	1.00 (reference)		1.00 (reference)		1.00 (reference)	
Q2	0.86 (0.79–0.92)	0.000	0.88 (0.82–0.95)	0.001	0.97 (0.91–1.04)	0.380
Q3	0.71 (0.65–0.77)	0.000	0.74 (0.68–0.80)	0.000	0.90 (0.83–0.97)	0.006
Age	-	-	0.98 (0.97–0.98)	0.000	0.95 (0.94–0.96)	<0.001
Sex (Male)	-	-	1.41 (1.32–1.50)	0.000	1.54 (1.45–1.63)	<0.001
Diabetes	-	-	-	-	1.28 (1.20–1.37)	<0.001
Hypertension	-	-	-	-	1.38 (1.26–1.50)	<0.001
Chronic Kidney Disease	-	-	-	-	0.99 (0.99–1.00)	<0.001
Deficit Accumulation Frailty Index Not Frail Pre-Frail Frail	-	-	-	-	1.00 (reference) 2.09 (1.94–2.25) 2.59 (2.36–2.83)	<0.001 <0.001
Education Completion <12 years 12 years ≥13 years	-	-	-	-	1.00 (reference) 0.89 (0.80–0.98) 0.83 (0.78–0.89)	0.017 <0.001

MASLD: Metabolic dysfunction-associated steatotic liver disease; FLI: Fatty Liver Index; Fully adjusted model includes adjustments for: age, sex, diabetes, hypertension, chronic kidney disease, frailty status, and education status.

**Table 3 nutrients-17-01415-t003:** MASLD + Ultra Processed Food (Poisson Regression).

MASLD (FLI ≥ 60) vs. No-MASLD (FLI < 30)						
	Unadjusted/Crude Model Relative Risk (95% CI)	*p*-Value	Age and Sex Adjusted Model Relative Risk (95% CI)	*p*-Value	Fully Adjusted Model Relative Risk (95% CI)	*p*-Value
Ultra Processed Food Score (UPF)						
Q1	1.00 (reference)		1.00 (reference)		1.00 (reference)	
Q2	1.16 (1.07–1.27)	0.000	1.13 (1.04–1.23)	0.003	1.12 (1.04–1.21)	0.002
Q3	1.25 (1.16–1.36)	0.000	1.16 (1.07–1.26)	0.000	1.13 (1.05–1.22)	0.001
Age	-	-	0.97 (0.97–0.98)	0.000	0.95 (0.94–0.96)	<0.001
Sex (Male)	-	-	1.42 (1.33–1.52)	0.000	1.53 (1.44–1.62)	<0.001
Diabetes	-	-	-	-	1.30 (1.12–1.39)	<0.001
Hypertension	-	-	-	-	1.39 (1.27–1.51)	<0.001
Chronic Kidney Disease	-	-	-	-	0.99 (0.99–0.99)	<0.001
Deficit Accumulation Frailty Index Not Frail Pre-Frail Frail	-	-	-	-	1.00 (reference) 2.07 (1.92–2.23) 2.56 (2.35–2.82)	<0.001 <0001
Education Completion <12 years 12 years ≥13 years	-	-	-	-	1.00 (reference) 0.87 (0.79–0.96) 0.82 (0.77–0.88)	0.006 <0.001

MASLD: Metabolic dysfunction-associated steatotic liver disease; FLI: Fatty Liver Index; Fully adjusted model includes adjustments for: age, sex, diabetes, hypertension, chronic kidney disease, frailty status, and education status.

**Table 4 nutrients-17-01415-t004:** MASLD + Mediterranean Diet + Ultra Processed Food (Poisson Regression).

MASLD (FLI ≥ 60) vs. No-MASLD (FLI < 30)	Fully Adjusted Model Relative Risk (95% CI)	*p*-Value
Mediterranean Diet		
Q1	1.00 (reference)	
Q2	0.96 (0.90–1.02)	0.251
Q3	0.88 (0.82–0.96)	0.002
Ultra Processed Foods		
Q1	1.00 (reference)	
Q2	1.13 (1.05–1.21)	0.001
Q3	1.14 (1.06–1.23)	0.001
Age	0.95 (0.94–0.96)	<0.001
Sex (Male)	1.50 (1.41–1.60)	<0.001
Diabetes	1.30 (1.22–1.39)	<0.001
Hypertension	1.38 (1.27–1.51)	<0.001
Chronic Kidney Disease	0.99 (0.99–0.99)	<0.001
Deficit Accumulation Frailty Index Not Frail Pre-Frail Frail	1.00 (reference) 2.06 (1.90–2.22) 2.53 (2.31–2.78)	<0.001 <0.001
Education Completion <12 years 12 years ≥13 years	1.00 (reference) 0.88 (0.79–0.97) 0.83 (0.78–0.89)	0.010 <0.001

MASLD: Metabolic dysfunction-associated steatotic liver disease; FLI: Fatty Liver Index; Fully adjusted model includes adjustments for: age, sex, diabetes, hypertension, chronic kidney disease, frailty status, and education status.

**Table 5 nutrients-17-01415-t005:** MASLD vs. Indeterminate/No-MASLD + Mediterranean Diet (Poisson Regression).

MASLD (FLI ≥ 60) vs. Indeterminate and No-MASLD (FLI < 60)	Fully Adjusted Model Relative Risk (95% CI)	*p*-Value
Mediterranean Diet		
Q1	1.00 (reference)	
Q2	0.94 (0.87–1.02)	0.137
Q3	0.90 (0.81–0.97)	0.014
Age	0.95 (0.94–0.95)	<0.001
Sex (Male)	1.42 (1.33–1.53)	<0.001
Diabetes	1.43 (1.33–1.53)	<0.001
Hypertension	1.36 (1.23–1.50)	<0.001
Chronic Kidney Disease	0.99 (0.99–1.00)	<0.001
Deficit Accumulation Frailty Index Not Frail Pre-Frail Frail	1.00 (reference) 2.33 (2.14–2.54) 3.09 (2.78–3.43)	<0.001 <0.001
Education Completion <12 years 12 years ≥13 years	1.00 (reference) 0.89 (0.79–1.00) 0.82 (0.76–0.89)	0.054 <0.001

MASLD: Metabolic dysfunction-associated steatotic liver disease. FLI: Fatty Liver Index. Fully adjusted model includes adjustments for: age, sex, diabetes, hypertension, chronic kidney disease, frailty status, and education status.

**Table 6 nutrients-17-01415-t006:** MASLD vs. Indeterminate/No-MASLD + Ultra-processed Food (Poisson Regression).

MASLD (FLI ≥ 60) vs. Indeterminate and No-MASLD (FLI < 60)	Fully Adjusted Model Relative Risk (95% CI)	*p*-Value
Ultra Processed Foods		
Q1	1.00 (reference)	
Q2	1.17 (1.07–1.27)	0.001
Q3	1.18 (1.08–1.29)	<0.001
Age	0.94 (0.94–0.95)	<0.001
Sex (Male)	1.41 (1.31–1.51)	<0.001
Diabetes	1.45 (1.34–1.57)	<0.001
Hypertension	1.36 (1.24–1.51)	<0.001
Chronic Kidney Disease	0.99 (0.99–0.99)	<0.001
Deficit Accumulation Frailty Index Not frail Pre-Frail Frail	1.00 (reference) 2.32 (2.13–2.53) 3.07 (2.76–3.42)	<0.001 <0.001
Education Completion <12 years 12 years ≥13 years	1.00 (reference) 0.87 (0.78–0.98) 0.81 (0.75–0.87)	0.022 < 0.001

MASLD: Metabolic dysfunction-associated steatotic liver disease; FLI: Fatty Liver Index; Fully adjusted model includes adjustments for: age, sex, diabetes, hypertension, chronic kidney disease, frailty status, and education status.

**Table 7 nutrients-17-01415-t007:** Unadjusted “Heat Map” of rates by MASLD by tertile.

% of MASLD (≥60 vs. <60)	UPF Tertile 1	UPF Tertile 2	UPF Tertile 3
MedDiet Tertile 1	29.45%	34.57%	38.18%
MedDiet Tertile 2	24.97%	30.61%	30.36%
MedDiet Tertile 3	21.66%	25.55%	27.01%

MASLD: Metabolic dysfunction-associated steatotic liver disease; UPF: Ultra Processed Food. MedDiet: Mediterranean Diet. Colour gradation represents risk, with green as the lowest risk and red as the highest.

**Table 8 nutrients-17-01415-t008:** Linear Regression Mediterranean Diet.

	Age and Sex Adjusted Model: Coefficient (95% CI)	*p*-Value
**Triglycerides**		
Mediterranean Diet		
Q1	1.00 (reference)	
Q2	−6.30 (−9.29–−3.30)	<0.001
Q3	−10.27 (−13.30–−7.24)	<0.001
Age	−0.28 (−0.58–0.02)	0.065
Sex (Male)	−5.98 (−8.46–−3.50)	<0.001
**Body Mass Index**		
Mediterranean Diet		
Q1	1.00 (reference)	
Q2	−0.24 (−0.49–0.01)	0.064
Q3	−0.72 (−0.97–−0.46)	<0.001
Age	−0.12 (−0.14–−0.09)	<0.001
Sex (Male)	−0.29 (−0.50–−0.08)	0.006
**Waist Circumference**		
Mediterranean Diet		
Q1	1.00 (reference)	
Q2	−0.74 (−1.40–−0.08)	0.029
Q3	−2.05 (−2.73–−2.39)	<0.001
Age	−0.16 (−0.22–−0.09)	<0.001
Sex (Male)	8.13 (7.59–8.67)	<0.001
**Gamma glutamyl-transferase (GGT)**		
Mediterranean Diet		
Q1	1.00 (reference)	
Q2	−1.25 (−2.92–0.42)	0.142
Q3	−1.09 (−2.78–0.59)	0.203
Age	0.06 (−0.10–0.23)	0.46
Sex (Male)	4.22 (2.84–5.60)	<0.001

**Table 9 nutrients-17-01415-t009:** Linear Regression Ultra Processed Foods.

	Age and Sex Adjusted Model: Coefficient (95% CI)	*p*-Value
**Triglycerides**		
Ultra Processed Foods		
Q1	1.00 (reference)	
Q2	4.47 (1.44–7.51)	0.004
Q3	4.21 (1.13–7.29)	0.007
Age	−0.18 (−0.49–0.12)	0.240
Sex (Male)	−5.47 (−8.00–−2.93)	<0.001
**Body Mass Index**		
Ultra Processed Foods		
Q1	1.00 (reference)	
Q2	0.37 (0.12–0.63)	0.004
Q3	0.36 (0.10–0.62)	0.006
Age	−0.12 (−0.14–−0.09)	<0.001
Sex (Male)	−0.27 (−0.48–−0.05)	0.014
**Waist Circumference**		
Ultra Processed Foods		
Q1	1.00 (reference)	
Q2	0.95 (0.29–1.62)	0.005
Q3	1.07 (0.39–1.75)	0.002
Age	−0.15 (−0.21–0.08)	<0.001
Sex (Male)	8.18 (7.62–8.73)	<0.001
**Gamma glutamyl-transferase (GGT)**		
Ultra Processed Foods		
Q1	1.00 (reference)	
Q2	−0.62 (−2.30–1.06)	0.467
Q3	1.04 (−0.66–2.75)	0.230
Age	0.06 (−0.11–0.23)	0.471
Sex (Male)	4.23 (2.82–5.63)	<0.001

## Data Availability

The original contributions presented in this study are included in the article/Appendix A, further inquiries can be directed to the corresponding author/s.

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
