# Peer review of "Associations Between MASLD, Ultra-Processed Food and a Mediterranean Dietary Pattern in Older Adults"

_nutrients, 2025, doi:10.3390/nu17091415_

Round 1
Reviewer 1 Report
Comments and Suggestions for Authors
The study design is a bit complicated. I understand that ALSOP study started later than the first study and there's a lag in blood sample collection. You've described an algorithm for using the data, but it doesn't seem right to me. In the case where a person doesn't have initial data, did you use the data they had, regardless of the time point in which it was obtained? This isn't clear to me; if so, it wouldn't be correct. It would be more appropriate not to use that data, wouldn't it?
The Mediterranean diet and ultra-processed food scores are not described. I would like to know how the authors calculated these indexes, what foods were taken into account. There is not any description of the frailty index as well, and I would like to know how was calculated and what the frailty index consisted of.
Author Response
REVIEWER 1
The study design is a bit complicated. I understand that ALSOP study started later than the first study and there's a lag in blood sample collection. You've described an algorithm for using the data, but it doesn't seem right to me. In the case where a person doesn't have initial data, did you use the data they had, regardless of the time point in which it was obtained? This isn't clear to me; if so, it wouldn't be correct. It would be more appropriate not to use that data, wouldn't it?
The Mediterranean diet and ultra-processed food scores are not described. I would like to know how the authors calculated these indexes, what foods were taken into account. There is not any description of the frailty index as well, and I would like to know how was calculated and what the frailty index consisted of.
Thank you for these comments and we appreciate there is considerable information in this study to digest. Firstly, regarding the score calculation and missing data: no, if there was no relevant data available we did not use the data. We have added a sentence to the methods section 2.3 which reads: “If there was either no GGT data available or if the other anthropometric or triglyceride data was more than 1 year from the GGT collection, the FLI was not calculated, and the individual was not evaluated in this study.”
Secondly, regarding the Mediterranean Diet and UPF scores, we had cited the paper that described their derivation in detail (see our previous reference #37, doi:10.3390/nu16172978). To make this easier for readers of this paper, we’ve added a supplement which includes the overall food frequency questionnaire found in ALSOP as well as the score derivation methodology.
Finally, in terms of the frailty index, this has been published on previously (see our previous reference #47, doi:10.1093/gerona/glab225); to make this clearer we have added the text: “The deficit accumulation frailty index is a 67-item score generated yearly in ASPREE, based on the principles of Rockwood et al. [48]. More details can be seen in the derivation and validation paper published in 2022 [47].”
Describing this in great detail is beyond the scope of this paper but those interested will be able to find the methodology described clearly in these references.
Reviewer 2 Report
Comments and Suggestions for Authors
The manuscript presents the results of a study investigating the association between dietary patterns - specifically Mediterranean diet and ultra-processed foods (UPFs) - and the risk of metabolic dysfunction-associated steatotic liver disease (MASLD) in older Australians. Using data from 6,753 community-dwelling participants aged ≥70 years from the ASPREE cohort study, dietary scores (ASPREE-MDS and UPF index) and the Fatty Liver Index (FLI) were analysed. Greater adherence to the Mediterranean diet was consistently associated with a lower risk of MASLD, even after adjusting for age, sex, and comorbidities. Conversely, higher UPF intake significantly increased the risk of MASLD. Both dietary patterns independently and jointly influenced the FLI and its components, with adherence to the Mediterranean diet linked to lower BMI, waist circumference, and triglyceride levels, while a higher UPF intake demonstrated the opposite trend. The analysis of the collected data enabled the identification of significant associations regarding the prevalence of steatotic liver disease in relatively healthy older adults. A higher intake of the Mediterranean diet was associated with a reduced risk of MASLD. Furthermore, individuals with simultaneously high adherence to both the Mediterranean diet and consumption of ultra-processed foods had a lower incidence of MASLD. The manuscript is interesting, and the presentation of results is concise and logically structured. The results obtained highlight a major benefit of the Mediterranean diet - namely, the reduced incidence of steatohepatitis associated with metabolic dysfunction.
Minor revision:
Tables 5 and 6 must be presented in format portrait and not in landscape format.
Author Response
REVIEWER 2
The manuscript presents the results of a study investigating the association between dietary patterns - specifically Mediterranean diet and ultra-processed foods (UPFs) - and the risk of metabolic dysfunction-associated steatotic liver disease (MASLD) in older Australians. Using data from 6,753 community-dwelling participants aged ≥70 years from the ASPREE cohort study, dietary scores (ASPREE-MDS and UPF index) and the Fatty Liver Index (FLI) were analysed. Greater adherence to the Mediterranean diet was consistently associated with a lower risk of MASLD, even after adjusting for age, sex, and comorbidities. Conversely, higher UPF intake significantly increased the risk of MASLD. Both dietary patterns independently and jointly influenced the FLI and its components, with adherence to the Mediterranean diet linked to lower BMI, waist circumference, and triglyceride levels, while a higher UPF intake demonstrated the opposite trend. The analysis of the collected data enabled the identification of significant associations regarding the prevalence of steatotic liver disease in relatively healthy older adults. A higher intake of the Mediterranean diet was associated with a reduced risk of MASLD. Furthermore, individuals with simultaneously high adherence to both the Mediterranean diet and consumption of ultra-processed foods had a lower incidence of MASLD. The manuscript is interesting, and the presentation of results is concise and logically structured. The results obtained highlight a major benefit of the Mediterranean diet - namely, the reduced incidence of steatohepatitis associated with metabolic dysfunction. For this reason, I recommend the publication of this manuscript.
Thank you for your support and kind words regarding our work
Minor revision:
Tables 5 and 6 must be presented in format portrait and not in landscape format.
Thank you – this change has been made, though may require additional formatting from the Nutrients team.
Reviewer 3 Report
Comments and Suggestions for Authors
This study presents valuable insights into the associations between MASLD, ultra-processed food consumption, and adherence to a Mediterranean dietary pattern in older adults. Its rigorous methodology and integration with existing literature enrich understanding of dietary impacts on liver health within this demographic. The analysis of dietary modalities provides a comprehensive overview of the clinical implications and potential public health recommendations. However, some points merit further consideration:
1.The study predominantly utilized the Fatty Liver Index (FLI) for MASLD identification, which, while validated for epidemiological studies, is less precise compared to imaging techniques or biopsy. Could employing additional diagnostic modalities bolster the reliability of findings and their applicability to clinical practice?
2.The use of tertiles for dietary scoring is effective for internal population comparisons but limits extrapolation to broader epidemiological frameworks. Would incorporating more granular data on food quantities or caloric intake enhance the precision and relevance of dietary associations?
3.The interplay between Mediterranean dietary adherence and ultra-processed food intake is highlighted, yet the mechanisms underlying the observed protective effects of the Mediterranean diet remain speculative. Could additional biomarker analyses or dietary component-specific investigations elucidate these pathways more comprehensively?
4.While the study notes socio-economic barriers to dietary adherence, these factors are discussed primarily in broad terms. Would a deeper exploration of specific drivers behind ultra-processed food consumption in older adults, such as convenience or social isolation, provide actionable insights for tailored public health interventions?
5.The predominantly Caucasian population of the ASPREE cohort may limit the generalizability of findings to other ethnic groups. Could expanding the study to include diverse populations improve the robustness of conclusions and their applicability in multicultural settings?
6.Food recall data are inherently prone to bias, particularly when self-reported over extended periods. Would employing objective measures such as dietary biomarkers enhance data accuracy and mitigate recall-related limitations?
Author Response
REVIEWER 3
This study presents valuable insights into the associations between MASLD, ultra-processed food consumption, and adherence to a Mediterranean dietary pattern in older adults. Its rigorous methodology and integration with existing literature enrich understanding of dietary impacts on liver health within this demographic. The analysis of dietary modalities provides a comprehensive overview of the clinical implications and potential public health recommendations. However, some points merit further consideration:
1. The study predominantly utilized the Fatty Liver Index (FLI) for MASLD identification, which, while validated for epidemiological studies, is less precise compared to imaging techniques or biopsy. Could employing additional diagnostic modalities bolster the reliability of findings and their applicability to clinical practice?
Yes – we were limited to what was available in this large and rich epidemiological dataset of older Australian adults, but certainly radiology or biopsy would be perhaps better (despite, as has been noted here, the validation of the FLI for epidemiological work). We have accordingly made a slight amendment to the ‘limitations’ section to read: “The use of FLI to identify those with MASLD is a limitation in the diagnosis of steatotic liver disease rather than the gold-standard of liver biopsy or radiology.”
2.The use of tertiles for dietary scoring is effective for internal population comparisons but limits extrapolation to broader epidemiological frameworks. Would incorporating more granular data on food quantities or caloric intake enhance the precision and relevance of dietary associations?
Certainly – if there was the opportunity to incorporate granular food quantities or caloric intake it would be an additional and valuable nuance to our work, but unfortunately as can be seen in the new supplementary data, this was unable to be readily estimated from the ALSOP dietary questionnaire.
This has been highlighted in the limitations section here: “Given that the scores were derived from the ASPREE dietary questionnaire, and not a validated food frequency questionnaire, we did not have information on the quantities of food eaten, only the frequencies. This limited our ability to apply and compare with previously used epidemiological dietary scores, as well as limiting our understanding of whether there are threshold quantities of important dietary components for either benefit or risk or are related to total calorie intake. Furthermore, the ASPREE-dietary questionnaire was self-reported, based on the previous 12 months of food intake, potentially leading to recall bias.”
3.The interplay between Mediterranean dietary adherence and ultra-processed food intake is highlighted, yet the mechanisms underlying the observed protective effects of the Mediterranean diet remain speculative. Could additional biomarker analyses or dietary component-specific investigations elucidate these pathways more comprehensively?
Yes – this is an interesting point, and one we’ve added to our limitations section: “An additional limitation inherent to this study design is the lack of potential biomarker discovery to evaluate any specific protective factors found in the Mediterranean Diet for those also eating significant quantities of UPFs; future biomarker-driven mechanistic work would be valuable.”
4.While the study notes socio-economic barriers to dietary adherence, these factors are discussed primarily in broad terms. Would a deeper exploration of specific drivers behind ultra-processed food consumption in older adults, such as convenience or social isolation, provide actionable insights for tailored public health interventions?
This is an incredibly important point you have raised and is – we believe – worthy of its own article. We believe that evaluating potential barriers – including location (for example, urban vs regional/rural), sociodemography/financial situation, and support at home would be extremely valuable. We have highlighted this in the discussion here: “Social and economic drivers of UPF intake in older adults warrants further deep exploration to allow targeted public health messaging to convey the importance of healthy dietary patterns.”
5.The predominantly Caucasian population of the ASPREE cohort may limit the generalizability of findings to other ethnic groups. Could expanding the study to include diverse populations improve the robustness of conclusions and their applicability in multicultural settings?
Certainly; unfortunately, we are limited to those included in this pre-existing large epidemiological set. It’s likely that some of the barriers and food choices beyond Australia and indeed beyond Caucasian/White older persons will differ, and comparative studies across geographies would be valuable. We have included a point to this effect in the limitations section: “Finally, this population was predominantly Caucasian, limiting the ability to apply these findings to other ethnicities.”
6.Food recall data are inherently prone to bias, particularly when self-reported over extended periods. Would employing objective measures such as dietary biomarkers enhance data accuracy and mitigate recall-related limitations?
It’s possible that specific biomarkers would help to reduce the risk of this – and we’ve added a point about biomarkers in response to your interesting query #3. On the other hand, biomarkers are necessarily somewhat limited in comparison to whole-diet eating patterns; capturing the heterogeneity of intake with only a few biomarkers has its own limitations, and so we believe that a combination of recall and key biomarkers would be the most valuable approach, but unfortunately one we are limited in here due to the lack of available biomarkers in this dataset.
Thank you again for your detailed queries and engagement with our work.
Reviewer 4 Report
Comments and Suggestions for Authors
In the present study, Isabella Commins and co-workers aimed to examine the impact of the Mediterranean Diet and Ultra Processed Food (UPF) intake on the risk of prevalent Metabolic Dysfunction-Associated Steatotic Liver Disease (MASLD) in older Australian adults. The authors on the basis of present results suggest that a public health messaging should focus on promoting Mediterranean dietary habits, and ways to help older people achieve this given the social and economic barriers they may face.
Overall, I think the paper is timely, very interesting (within the scope of “Nutrients”) and the data are relevant, particularly in the field of geriatric nutrition. In my humble opinion, I make some suggestions for further improving the quality of manuscript.
1) Please better justify the sample size of the present study, in relation to the expected magnitude of effects of the Mediterranean Diet and UPF intake in Australian older adults.
2) Isoflavones could be present at higher concentrations in legumes and/or in Mediterranean diet “modified” (similar to “Oriental diet”). So, the positive effects on the outcome measures here described could be related to these phytochemicals. Please discuss this aspect in the revised manuscript considering for your convenience these references (J. Clin. Endocrinol. Metab. 2013, 98, 3366-3374; Circulation. 2020, 141, 1127-1137; Nutrients, 2022, 14, 1550).
3) Some medicines/drugs can be used to treat the enrolled patients. This aspect could interfere with the results here revealed. Please discuss this aspect and eventually take this factor into account in data analysis.
4) Recent research suggested that gut microbiome composition is related to the risk of developing MASLD in older adults (see Nat Rev Gastroenterol Hepatol 2020, 17, 279-297). Do the authors plan to consider microbioma composition in this population? Please make a careful comment in the discussion section of manuscript.
5) In light of the results here obtained, please discuss the possible application of nutraceutics and/or functional foods that, in combination with Mediterranean diet and physical activity could provide a possible further strategy to ameliorate metabolic health risk factors and/or MASLD pattern in older adults (Nutrients, 2022, 14, 1550).
Author Response
REVIEWER 4
In the present study, Isabella Commins and co-workers aimed to examine the impact of the Mediterranean Diet and Ultra Processed Food (UPF) intake on the risk of prevalent Metabolic Dysfunction-Associated Steatotic Liver Disease (MASLD) in older Australian adults. The authors on the basis of present results suggest that a public health messaging should focus on promoting Mediterranean dietary habits, and ways to help older people achieve this given the social and economic barriers they may face.
Overall, I think the paper is timely, very interesting (within the scope of “Nutrients”) and the data are relevant, particularly in the field of geriatric nutrition. In my humble opinion, I make some suggestions for further improving the quality of manuscript.
Thank you for your kind words and the time you’ve taken to allow us to improve our manuscript.
1) Please better justify the sample size of the present study, in relation to the expected magnitude of effects of the Mediterranean Diet and UPF intake in Australian older adults.
We were necessarily ‘limited’ in our sample size selection due to the pre-existing nature of the established clinical trial-based dataset, and thus a power calculation was not performed – not least because of the relative lack of work in this space specifically. We believe that – given the statistical and clinical significance of the results – the dataset size and power associated with this study was adequate. We are not sure how you’d specifically like us to address this in the manuscript, but would be happy to include if you have a suggested phrase.
2) Isoflavones could be present at higher concentrations in legumes and/or in Mediterranean diet “modified” (similar to “Oriental diet”). So, the positive effects on the outcome measures here described could be related to these phytochemicals. Please discuss this aspect in the revised manuscript considering for your convenience these references (J. Clin. Endocrinol. Metab. 2013, 98, 3366-3374; Circulation. 2020, 141, 1127-1137; Nutrients, 2022, 14, 1550).
Thank you for this interesting point. We have added a section to the discussion to address the potential protective benefit of isoflavones and other components of the Mediterranean Diet, which reads as follows and includes those key references:
“Mechanistically, there are multiple known phytochemicals found in our food which may influence the development of steatosis and ameliorate dyslipidemia, including isoflavones [51-53] (often found in legumes), the polyphenols found in extra virgin olive oil [54,55], and other bioactive compounds found in many of the plant-based foods commonly consumed in the Mediterranean Diet [56].”
3) Some medicines/drugs can be used to treat the enrolled patients. This aspect could interfere with the results here revealed. Please discuss this aspect and eventually take this factor into account in data analysis.
While various medications can be used for the cardiometabolic conditions associated with MASLD, they will be over-represented in the MASLD group and, in turn, likely reduced the prevalence of MASLD amongst the otherwise at-risk population. Given this, their inclusion is likely to have either no significance or to reduce the significance of our results in this instance, particularly given that many are already implicitly included in the categorical modifiers of our statistics (i.e., the definition of ‘diabetes’ and ‘hypertension’ includes the use of antidiabetic medication(s) and antihypertensive(s) respectively). Due to this, and as the focus in this study is on diet, we would prefer to not expand the paper with multiple additional pharmaceutical related analyses which are unlikely to change the significance of our results.
4) Recent research suggested that gut microbiome composition is related to the risk of developing MASLD in older adults (see Nat Rev Gastroenterol Hepatol 2020, 17, 279-297). Do the authors plan to consider microbiome composition in this population? Please make a careful comment in the discussion section of manuscript.
We have added a segment following the isoflavone and phytochemical section of the discussion linking the Mediterranean Diet to improved microbiotal diversity and, in turn, an ‘improved’ microbiome to reduced MASLD:
“Similarly, the Mediterranean Diet may have beneficial effects on the gut microbiome [60], which is potentially causative (or protective) for the development of MASLD [61].”
5) In light of the results here obtained, please discuss the possible application of nutraceutics and/or functional foods that, in combination with Mediterranean diet and physical activity could provide a possible further strategy to ameliorate metabolic health risk factors and/or MASLD pattern in older adults (Nutrients, 2022, 14, 1550).
We thank you for this consideration – however, we do note that as we don’t have any access to biochemical parameters or dietary quantities of isoflavones, polyphenols, etc., as well as no direct neutraceutical supplementation in this group, it would be quite speculative to link potential benefits in our discussion and not directly linked to the main purposes of this manuscript. While we do believe that discussing the potential benefits of specific food micro-constituents (as above, for comment #2), promoting the use of neutraceuticals in the setting of our findings does seem premature given the lack of granular data on these compounds we have available in our dataset.
Thank you again for taking the time to review our manuscript.